# Titanium Oxide Microspheres with Tunable Size and Phase Composition

**DOI:** 10.3390/ma12091472

**Published:** 2019-05-07

**Authors:** Anton S. Poluboyarinov, Vitaly I. Chelpanov, Vasily A. Lebedev, Daniil A. Kozlov, Kristina M. Khazova, Dmitry S. Volkov, Irina V. Kolesnik, Alexey V. Garshev

**Affiliations:** 1Faculty of Materials Science, Lomonosov Moscow State University, Leninskiye Hills 1, Moscow 119234, Russia; anpolvk@gmail.com (A.S.P.); danilkozlove@gmail.com (D.A.K.); khazovakm@gmail.com (K.M.K.); kolesnik.iv@gmail.com (I.V.K.); 2Baikov Institute of Metallurgy and Material Science RAS, Leninsky Avenue 49, Moscow 119334, Russia; vitaliy.chelpanov@gmail.com; 3Faculty of Chemistry, Lomonosov Moscow State University, Leninskiye Hills 1, Moscow 119234, Russia; vasya_lebedev@mail.ru (V.A.L.); dmsvolkov@gmail.com (D.S.V.); 4Institute of General Inorganic Chemistry RAS, Leninsky Avenue 31, Moscow 119071, Russia

**Keywords:** titania microspheres, sol-gel synthesis, amorphous titania, size control, alkoxide hydrolysis, optical absorption edge, SPF

## Abstract

Due to their unique physical and chemical properties, monodisperse titanium oxide microspheres can be used in dye-sensitized solar cells, as cosmetic pigments, and for other applications. However, the synthesis of microspheres with narrow size distribution, desired phase composition, and porosity is still a challenge. In this work, spherical titania particles with controllable size, crystallinity, and pore size were obtained by Ti(O^n^Bu)_4_ hydrolysis in ethanol. The influence of NaOH addition on the particles’ size and morphology was investigated for the first time. Particle diameter can be tailored from 300 nm to 1.5 μm by changing water and NaOH concentrations. Particle size was analyzed by the statistical processing of scanning electron microscopy (SEM) images and differential centrifugal sedimentation (DCS) measurements. Optical properties of the microspheres were studied by diffuse reflectance UV-Vis spectroscopy. Thermal and hydrothermal treatment allowed transforming amorphous phase in as-prepared particles into nanocrystalline anatase and/or rutile. Transmission electron microscopy (TEM) study of the lamellae, cut out from spherical particles using focused ion beam (FIB), revealed that as-synthesized microspheres are non-hollow, homogeneous, and crystallize throughout the whole volume of the particle. The spherical particles possess photoprotective properties; the highest sun protection factor (SPF) was observed for amorphous microspheres.

## 1. Introduction

TiO_2_ particles can be used as cosmetic pigment [1,2,3], a component of dye-sensitized solar cells [4,5,6,7], sorbent material for high-performance liquid chromatography (HPLC) [8,9,10,11], and other applications. The spherical morphology of the particles is preferable for all mentioned applications, while the requirements for other properties including the crystallinity, pore size, and particle size can differ. Cosmetic pigments require particles of micrometer-scale size and low crystallinity, because amorphous titanium oxide does not possess high photocatalytic activity [12], and therefore does not serve as a source of free radicals, and does not cause oxidative stress. Spherical particles provide high scattering, which is important for the soft focus effect [13]. In contrast to cosmetic applications, such applications as water purification require high photocatalytic activity [14,15,16,17,18]. Particles for chromatography and radionuclide treatment are expected to have spherical shape [19] and narrow pore size distribution in order to minimize flow resistance. Due to the pressure limitations under HPLC conditions, the size of the sorbent should be of micrometer scale. Apart from these factors, it is optimal to use particles that have a pore size of 5–50 nm, and consist of crystalline phases to ensure the required adsorption capacity and chemical stability, and TiO_2_ microspheres have shown good results in sorption applications [15,20]. For solar cells applications, a high degree of crystallinity is important to avoid the recombination of photo-generated charge carriers. Moreover, for some types of cells, the usage of 0.4–0.6 spherical particles is presumed to be optimal due to their scattering effects [21].

The sol–gel route allows one to obtain amorphous spherical particles, which can later be crystallized to obtain the desired phase composition. Particles with diameters as small as 50 nm [22] and as large as 5 μm [8] were obtained. Although first attempts were polydisperse [23], over time, in some cases, remarkable monodispersity was achieved [24]. Colloidal stability plays an important role in particle growth: even agitation rate can be crucial for product morphology [25]. Three common approaches were developed to improve the reproducibility of the synthesis and prevent the aggregation of the particles: chemical modification [8,22,26,27,28], the addition of surfactants [24,29,30,31], and mineral additives, including but not limited to HCl [8,32], NaCl [30,32], KCl [30,33,34], and ammonia [24,35,36,37]. Chemical modification can be used to slow down hydrolysis and condensation reactions [38]. The influence of carboxylic acids with varying chain lengths was studied in [22]. Change of temperature [28,35] or a special agitation regime, such as Couette–Taylor flow [39], can be used as well.

This work investigates what can be achieved by adding NaOH and changing the amount of added water while using the common reaction of titanium n-butoxide hydrolysis in ethanol, although at a lowered temperature. Although poorly described in the literature, NaOH prevents aggregation and influences particle size. Initially, XRD amorphous particles can be later crystallized to obtain the desired phase composition; however, the shrinkage of the particles should be taken into account.

## 2. Materials and Methods

Deionized (DI) water (18 MΩ resistivity, Milli-Q, Merck Millipore, Darmstadt, Germany), titanium n-tetrabutoxide Ti(O^n^Bu)_4_ (97%, Aldrich 244112, St. Louis, MO, USA), and sodium hydroxide (NaOH, 99.99%, Aldrich 306576) were used as received. Ethanol was dried as described in [39].

The typical syntheses of amorphous titania microspheres were carried out in a 100-mL polypropylene flask. The reaction mixture volume was 30 mL, which corresponds to 150 mg of the final product. The selected volume of DI water and aqueous solution of NaOH were added to the absolute ethanol. The mixture was cooled down to −5 °C; then, titanium n-butoxide was added dropwise under vigorous stirring. Once all of the butoxide was added, stirring was continued for a minute, and then stopped. After the solution became cloudy, synthesis was continued for two more hours; then, cooling was stopped, and the solution was left to dry in a propylene beaker in air at room temperature for a week.

The induction time varied from less than a minute to several hours, depending on the concentrations of water and NaOH. Reproducibility of the synthesis has been found to be quite good. The highest difference in the particle mean diameter among the samples synthesized under the same conditions is in the range of the standard deviation. The scalability of the synthesis was investigated, and no differences in the particle size and morphology were observed at solution volumes at least up to 200 mL, which corresponds to approximately 1 g of product.

Total water content was calculated as the sum of added water, added NaOH solution, and residual water in the absolute alcohol. The titanium n-butoxide concentration was 58 mM for all samples. The hydrolysis ratio h was varied from 2 to 14, and is given as: (1)$$h=\frac{{c(H}_{2}O)}{{c(\mathrm{Ti}(O}^{n}{\mathrm{Bu})}_{4})}$$

After drying at room temperature samples, were washed with water and dried at 60 °C overnight to remove residual alcohol. Microspheres were annealed at heating rate of 1 °C/min without isothermal aging up to 400 °C, 500 °C, 625 °C, and 700 °C. Crystallization temperatures were chosen with respect to the high-temperature powder X-ray diffraction (HTXRD) data. Samples for the measurements of photoprotective properties were prepared as follows: heating rate in air with 5 °C/min dwelling at 700 °C or 950 °C for anatase and rutile crystallization, respectively, and quenching. Additionally, hydrothermal treatment was used. For that, as-prepared microspheres (200 mg) were added to DI water (30 mL), and the suspension was transferred into the 60-mL Teflon-lined steel autoclave. The autoclave was held in a preheated oven at 200 °C for 24 h and 96 h; after that, the product was centrifuged (5000 rpm, 5 min), washed with water, and dried at 60 °C overnight.

X-ray diffraction (XRD) patterns were collected using a Rigaku D/MAX 2500 diffractometer (Tokyo, Japan) equipped with a rotating copper anode and a curved graphite monochromator. Experiments were carried out in θ/2θ Bragg–Brentano reflection geometry. High-temperature XRD experiments were performed with the use of the same diffractometer equipped with a high-temperature module. Temperature measurements were performed by a thermocouple placed directly on the platinum sample holder. Experiments were carried out in two different regimes: in air and under vacuum. Experiments were done in a stepped manner: the sample was heated at a rate of 5°C/min up to 1000 °C; every 50 °C, heating was stopped while an XRD measurement was conducted. The 2θ range was 5–70 with a step of 0.02 at a scan speed of 0.5 s per point. XRD data was processed using JANA2006 software [40]. The Rietveld method was used for profile analysis. The average size of the coherent scattering regions (CSR) was calculated from the Lorentzian contribution, $L_{x}$, to a pseudo-Voigt function, which was used for peak fitting. The content of amorphous phase was determined using the method, which is described in our previous work [12]; the absolute error was found to be approximately 5%.

The microstructure of the samples was characterized using a scanning electron microscope (SEM) Leo Supra 50VP. Detection was performed at an accelerating voltage of 20 kV. For particle size distribution (PSD), at least six images per sample were captured from randomly chosen locations and evaluated. Particle diameter was measured manually using an open-source software Gwyddion [41]. About 100 particles were measured per image, totaling 400–800 particles per sample.

Samples for transmission electron microscopy (TEM) were prepared with the use of an electron/ion scanning microscope CrossBeam 1540 EsB, Zeiss (Oberkochen, Germany). TEM and scanning transmission electron microscopy (STEM) image acquisition were done using a Fischione Model 3000 annular dark field detector and a Gatan Ultrascan 4 × 4k charge-coupled device (CCD) camera installed in a transmission electron microscope Zeiss Libra 200FE operated at 200 kV. Electron diffraction patterns were radially integrated in Gwyddion.

Thermogravimetric (TG) experiments were performed on a simultaneous thermogravimetric analyzer Netzsch STA 409 PC Luxx (Selb, Germany) with a heating rate of 10 °C up to 700 °C in air. Differential scanning calorimetry (DSC) data was obtained simultaneously. Evolved gases were analyzed by mass spectroscopy (MS).

Nitrogen sorption isotherms for multipoint Brunauer–Emmett–Teller (BET) analysis and Barrett–Joyner–Halenda (BJH) calculations were obtained at 77 K using a Quantachrome Nova 2400e instrument (Boynton Beach, FL, USA). Prior to the measurements, samples were evacuated for 3 h at 200 °C.

Differential centrifugal sedimentation (DCS) was performed on the differential disk centrifuge CPS Instruments DC24000 UHR. Sedimentation was stabilized by a density gradient within the fluid, and the accuracy of the measured sizes was ensured through the use of a calibration standard with a known particle size. Monodispersed polystyrene latex particles (Polymer Latex, Saint Petersburg, Russia) were used for size calibration. A refractive index of 1.627 at 405 nm was obtained by extrapolation of the data from the work [42]. The mixture of deionized water and ethanol (1:1 by volume) was used for the preparation of all suspensions. For amorphous titania particles, a refractive index value of 1.8 was used according to [43]. Full description of the measurement process can be found in the Appendix A.

Dynamic light scattering (DLS) measurements were performed on Zetasizer Nano ZS (Malvern instruments, Malvern, UK) equipped with 633-nm laser in backscatter detection mode. ζ-potential measurements were carried out after 24 h since the start of the synthesis. Four samples were investigated with values of *h* = 6, 12; and c(NaOH) = 0.5, 1.0 mM.

Inductively coupled plasma mass spectrometry (ICP-MS) measurements were performed on Elan DRC II (PerkinElmer, Waltham, MA, USA). More details about sample characterization by DCS, TEM, DLS, and ICP-MS are provided in the Appendix A.

Raman spectra were acquired using the Renishaw inVia Raman spectroscope coupled with a Leica DMLM optical microscope equipped with a 50× objective. Measurements were performed at room temperature in the Raman shift frequency range of 100 to 1460 cm^−1^ using a 50-mW 514-nm argon laser. Before the measurement, the Raman spectrometer was calibrated against the $F_{1g}$ line of Si at 520.2 cm^−1^ as a reference. The bands of rutile and anatase phases were assigned according to the literature [44].

UV-visible diffuse reflectance spectra were recorded on a Lambda 950 (PerkinElmer) spectrometer in range of 200 to 1000 nm in a diffuse reflectance regime.

To measure the sun protective factor (SPF), the sample was added to water-in-oil emulsion to make a 10 wt. % mixture. For more details about emulsion preparation, see the Appendix A. Then, 0.1 mL of this suspension was carefully distributed on the surface of the Vitro-Skin^®^ substrate. The spectra of the samples were recorded on a Lambda 950 spectrometer in the range of 290 to 400 nm (with an integrating sphere 150 mm in diameter), and the SPF was calculated according to ISO 24443 standard. For UV-illumination of the samples, a Suntest CPS + ATLAS MTS chamber was used. Commercially available cosmetic TiO_2_ pigment was used as a reference. The sample was studied by XRD and SEM. It consists of 77% rutile and 23% amorphous phase; the size of the particles is smaller than 200 nm.

## 3. Results and Discussion

### 3.1. Particle Morphology and Size

Products of titanium butoxide hydrolysis were investigated. At a low hydrolysis ratio (1) obtained particles are polydisperse (Figure 1a) due to the low reaction speed. At a higher hydrolysis ratio, the particles fuse together and form aggregates (Figure 1b), indicating a loss of colloidal stability. Therefore, the addition of a stabilizing agent is required.

A series of samples was prepared with varying NaOH concentrations (Figure 2a). At concentrations of NaOH 0.25 mM and less, samples consist of merged spherical-like particles. At higher NaOH concentrations (0.375–1.50 mM), samples consist of distinct spherical particles. According to the SEM data (Figure 3), the mean diameter of the as-synthesized particles decreases from 1.15 to 0.45 in the range 0.375 to 1.50 mM, while the relative standard deviation is within 9–14%. No changes were observed for samples in the range 1.0 to 1.5 mM. For c(NaOH) concentrations from 2.0 to 4.0 mM, obtained samples appear to consist of strongly fused particles.

Two series of samples with varying hydrolysis ratios (at c(NaOH) 0.625 and 1.25 mM) were synthesized. The addition of NaOH prevents fusing of the particles over the whole range of used hydrolysis ratios from 4 to 14 (Figure 2b). Upon the increase of the hydrolysis ratio, and correspondingly the hydrolysis rate, the particle diameter gradually and monotonically decreases (Figure 3). The same behavior was reported for n-butoxide hydrolysis in n-buthanol/acetonitril mixture in the presence of ammonia [45], but in our case, the broadening of the particle size distribution with the addition of water is not observed. At the same time, variations of water amounts in titanium ethoxide hydrolysis in ethanol in the presence of alkali metal chlorides were reported to show no significant effect on the size of the particles [30].

Thereby, distinct spherical particles with a relatively narrow size distribution can be obtained by direct titanium n-butoxide hydrolysis in the presence of NaOH. The final size of the obtained particles can be tuned from 300 up to 1500 nm by adjustments in $H_{2}$O or NaOH concentrations.

One of the disadvantages of evaluating particle size by SEM image processing is the locality of this method. In order to confirm its reproducibility, DCS was used as well. Not only is it suitable for micrometer-sized particles, it also enables one to evaluate PSD with remarkable resolution. As can be seen in Appendix A, particle sizes obtained from DCS are in a good agreement with the SEM data. DLS measurements were also performed, as this method is often used for spherical particles. All three methods give close results for the particles’ mean diameter and its deviation.

In addition to white precipitate, which consists of microspheres, monolithic yellowish solids (gel) are formed starting from a hydrolysis ratio of 8. When h is increased, the total amount of the spherical particles decreases, while the amount of monolithic gel increases. For 1.25-mM NaOH and hydrolysis ratios of 12 and 14, no microspheres were observed. Thus, particle sizes lower than 250 nm were not observed, because of the low stability of the primary particles in these conditions (Appendix A).

Titanium equilibrium solubility was measured using ICP-MS (Figure 4). Solubility increases with the increase of c(NaOH), but decreases with the increase of c(H_2_O). Under the observed conditions, 20% to 70% of titanium stays in solution, which is in agreement with [32,46]. The deposition of soluble Ti compounds during drying leads to the formation of monolithic gel, and is the cause for bad sample morphology at high c(NaOH). In the case of a high amount of water, low solubility gel is formed instead of spherical particles. Upon drying, the structure collapses, resulting in monolithic gel. If one decides to eliminate this stage, then changes in Ti yield (and most likely final particle size) should be taken into account.

Changes in morphology at low c(NaOH) indicate the colloidal stabilization of the particles. The most likely mechanism for such stabilization is the deprotonation of the surface –OH groups under alkali conditions. A decrease in particle size is affected both by an increase in colloidal stability and changes in solubility. At higher pH, more nuclei survive the early stages of the reaction (until stabilization is achieved due to an increase in the particle size), as their aggregation becomes less rapid. It is interesting to note that the lowering pH with HCl had the same effect [32]. Water in its turn causes the hydrolysis reaction to flow faster and at a greater degree, causing more nuclei to form and consequently to survive.

### 3.2. Processing

Obtained samples are XRD amorphous. The formation of anatase and transition from anatase to rutile are non-equilibrium processes. Its kinetics show a dependence on microstructure and sample history [47]. Samples prepared by different methods show different crystallization behavior.

#### 3.2.1. Choosing Optimal Conditions: High Temperature XRD

The crystallization kinetics of the samples were studied in two HTXRD experiments: in air and in vacuum. During the experiment in air, the first (titania) reflexes appear at 400 °C; they correspond to anatase phase (Figure 5). The transition to rutile phase starts at 750 °C and finishes at 950 °C, where only rutile reflexes are present (apart from platinum crucible). Therefore, three regions can be identified: anatase phase from 400 to 700 °C, an anatase–rutile mixture from 700 to 950 °C, and rutile from 950 °C. Pure anatase phase can be obtained with CSR ranging from 13 to 55 nm; see Figure 6. In the case of low peak intensities, CSR sizes are not shown due to large errors in such identification. Pure rutile phase can be obtained with CSR at about 150–200 nm. We can see that in the mixed region, the rutile CSR size starts from values higher than anatase.

In vacuum, anatase formation starts at 450 °C. The transition to rutile starts at 900 °C. Even at the highest observed temperature, the reflexes of the anatase phase are present. Therefore, two regions can be identified: pure anatase from 450 to 850 °C, and anatase–rutile composition from 850 °C. By extrapolation, we can assume that under these conditions, the anatase–rutile transition will end at 1100 °C.

Such temperatures of transition correspond to the surface crystallization mechanism [48]; that is, the generation of rutile nuclei takes place in the amorphous surface layer of anatase particles. A decrease of water partial pressure leads to a decrease in the mobility of the surface layers of anatase particles, leading to an increase of the rutile nuclei activation energy. Also in air, the measured rutile CSR size starts roughly from the anatase CSR size (Figure 6), indicating a possible transition of anatase particles into rutile. On the other hand, in vacuum, the rutile CSR size starts lower than that of anatase (at 900 °C 60 nm versus 120 nm), indicating the growth of new particles instead of the transition of the existing ones.

#### 3.2.2. Annealing: Phase Composition

Based on HTXRD data, the temperature range 400 to 700 °C was chosen for annealing at air. The samples were heated up to 400 °C, 500 °C, 625 °C, and 700 °C. In accordance with HTXRD study, anatase is the only phase present in the obtained samples (Appendix A). The BET results are given in Appendix A. With the increasing annealing temperature, the specific surface area changes linearly from 85 to 20 m^2^/g, and the pore diameter changes from 2.5 to 8 nm (Table 1).

Annealing at vacuum conditions (10^−2^ mbar) was performed, but it resulted in the carbonization of the residual organics. Additional investigation is required in order to obtain samples without residual carbon for cosmetic and chromatography applications.

#### 3.2.3. Annealing: Shrinkage

The shrinkage of the microspheres should be taken into account if one wants to obtain annealed particles with a predefined size. To get an upper bound of this effect, particles were also annealed at 1000 °C. As demonstrated in Figure 7, the particle mean diameter is reduced by 22% when heated up to 700 °C, and by 34% when heated up to 1000 °C. There are reports in the literature of diameter shrinkage values of 17% [27] and as high as 47% [28]. SEM analysis showed that the particles keep their spherical morphology, and no signs of sintering were observed up to 700 °C. For particles annealed at 1000 °C, sintering becomes notable (Figure 7d).

#### 3.2.4. Hydrothermal Treatment

Hydrothermal treatment was used as an alternative way for crystallization of the microspheres. Relatively rough conditions of 200 °C and 24 h were chosen. According to XRD data, anatase and brookite phases are present in a 4:1 ratio (Table 1). For the anatase phase, the crystallite size calculated from CSR was 18 nm; for brookite, it was not determined due to the low intensity of its reflexes. To measure the amorphous content, XRD study described in [12] was conducted. According to this study, the amount of amorphous titania content was calculated to be 55 ± 2%. Another sample was hydrothermally treated at the same temperature, but the duration was increased to 96 h. The anatase crystallite size did not change; it was calculated to be 21 nm. In this case, the amorphous content was calculated to be 42 ± 2%. TEM study of the sample hydrothermally treated at 200 °C for 24 h was conducted (Figure 8c,d). Anatase crystallites are present and distributed equally throughout the lamella. Thereby, crystallization takes place not only on the surface, but evenly throughout the volume of the particle.

Nitrogen low-temperature sorption experiments demonstrate that the microspheres stay porous after the treatment. The pore volume increases notably from 0.10 cm^3^/g to 0.33 cm^3^/g. It may be a result of the partial dissolution of the particles at hydrothermal conditions. Specific surface area values change from 220 m^2^/g to 115 m^2^/g when hydrothermally treated for 24 h and to 75 m^2^/g for 96 h. Average pore diameter increases after the treatment from 2.6 to 9.8 nm.

### 3.3. Internal Structure of the Microspheres

To investigate the internal structure of the as-synthesized particles, a TEM study was conducted. Lamellae about 100-nm thick were cut out of spherical particles using FIB. The TEM image and selected area electron diffraction (SAED) of a raw particle is shown in Figure 8a,b. In order to prevent contamination, particles were covered with several layers of Pt, which can be seen in the upper left corner of the image. No diffraction reflexes were observed on the SAED; hence, no crystalline inclusions are present in accordance with the XRD data. With this data, we can conclude that the as-synthesized particles are non-hollow and homogeneous.

The Raman spectrum of the as-prepared sample (Appendix A) contains several broad bands with complex profiles. The maxima of these bands appear more or less close to the positions that can correspond to anatase or rutile phases. The presence of the bands in the spectra of XRD-amorphous samples that are close to the bands of crystalline anatase and rutile indicates that the primary structure of amorphous phase comprises elements similar to crystalline phases. The spectrum of the sample annealed at 400 °C corresponds to anatase phase, and the spectrum of the sample annealed at 1000 °C contains the bands that correspond to rutile phase [44].

The results of the TG–DSC–MS analysis of the untreated sample are shown in Figure 9. The total weight loss is about 27% before 450 °C, and no additional weight loss is observed until 700 °C. Most of the weight loss occurs before 170 °C, which is illustrated as a single peak on the DTG curve at 120 °C. At the same position, there is an endothermic peak on DSC. MS data shows that the weight loss occurs due to dehydration, as there is a current of ions with a m/z ratio of 18, corresponding to water. Another feature is a small exothermic peak at 230 °C, which corresponds to dehydroxylation and/or the decomposition of organics. The second exothermic peak at 420 °C is usually associated with anatase formation from amorphous phase [24]. The XRD data are in good agreement with this theory, but since at such temperature significant CO_2_ elimination takes place, we should take into account the combustion of organics. In our previous work [12], we found out that partially amorphous titania without residual organics, such as Hombikat UV100, demonstrates no narrow exothermic peaks on DSC during crystallization. Hence, we can conclude that the main cause for the exothermic effect at 420 °C is the combustion of organic residuals.

According to the low-temperature nitrogen sorption data, the specific surface area of the amorphous particles equals 220 m^2^/g (Table 1). This result is in accordance with the literature data: the BET surface area of the sol–gel titania particles is usually as high as 200 to 300 m^2^/g [36].

### 3.4. Optical and Photoprotector Properties of the Microspheres

Optical properties, including the absorption edge and band-gap value, are important for using the microspheres in dye-sensitized solar cells. The optical absorption edge values of anatase-containing particles are close to each other (Figure 10a). The optical absorption edge of the amorphous phase is shifted to lower wavelengths, and the optical absorption edge of rutile is shifted to higher wavelengths.

The optical band gap (E_g_) of the microspheres was calculated from UV-Vis spectra according to the procedures suggested in [49,50,51]. As anatase and rutile are indirect semiconductors, the band gap was determined as an intercept of the linear part of the graph plotted in coordinates ${\left(F*h\nu \right)}^{\frac{1}{2}}$ versus hν (analog of Tauc plot), where hν is photon energy, and *F* is the Kubelka–Munk function, with an *x*-axis (Figure 10b). For diffuse reflectance:(2)$$F=\;\frac{{\left(1-R\right)}^{2}}{2R}$$
where *R* is reflectance [52]. The data are provided in Table 2. The band-gap value for the rutile-containing sample is 2.94 eV. This value is very close to the value 2.98 eV, which was published previously by Di Paola et al. [53]. For the two anatase-containing samples, the E_g_ values are 3.09 and 3.15 eV, whereas in many works, the optical band gap of anatase values vary from 3.05 to 3.2 eV [51,53,54,55]. Possible reasons for the optical band variation may be defects in anatase crystal lattice [56] and the presence of amorphous phase in the samples, which cannot be detected by conventional powder XRD.

The optical band gap for the amorphous phase was determined from diffuse reflectance spectra and equals 3.39 eV. This result is close to the optical band gap determined for anatase particles with a diameter of about 1 nm [55]. Monticone et al. [55] have shown that anatase particles with diameters smaller than 1.5 nm possess a quantum size effect and band-gap shift of 0.17 eV. In amorphous titanium oxide, the excitons may be confined in anatase or rutile-like fragments, which were detected by Raman spectroscopy (Appendix A).

Structural disorder defects may also have an influence on the optical properties of anatase and rutile microspheres in addition to the amorphous phase. Such defects may cause electronic defect states inside the band gap. They localize near the valence and/or conduction band edge and form tail-like energy levels (Urbach tails) [50,57]. It is clearly seen from Figure 10a that the amorphous phase has the longest tailing, whereas for the rutile-containing sample, it is the smallest. For solar cell, applications with rutile microspheres are more preferable than the others, because they contain less amorphous phase, and their absorption edge is shifted to higher wavelengths, providing a more effective harvesting of solar energy.

Titanium oxide is a well-known physical UV filter. The photoprotective properties of the samples have been tested for a series of the particles, which have a similar size and different phase composition (Table 3). To crystallize anatase or rutile, the samples were heated to 700 °C or 950 °C in air and quenched. The SPF for the amorphous microspheres is the highest among all the studied particles, and they are more attractive for cosmetic applications than crystalline microspheres.

## 4. Conclusions

In this work, we demonstrate that increasing the pH leads to particle stabilization, the lowering of particle size, and an increase of titanium solubility. Our experimental procedure allows one to obtain relatively monodisperse particles from 300 nm to 1.5 μm in diameter by adjusting water and NaOH concentrations. As-synthesized particles are amorphous, non-hollow, homogeneous, and contain residual water, ethanol, and n-butanol. The annealing atmosphere, which is presumably the partial pressure of water, has an effect on crystallization, and specifically on the anatase–rutile transition temperature. Crystallization of the particles is accompanied by an increase in pore size—calculated by BJH—of up to 8 nm in the case of annealing and to 10 nm for hydrothermal treatment. The shrinkage of the particles, which depends on treatment parameters, should be taken into account if particles of a specific size are desired. Hydrothermal treatment results into partial crystallization of the microspheres with up to 42% of titania being XRD amorphous, while shrinkage is not observed. The approach described in this paper can be applied to prepare titanium oxide microspheres with a tailored diameter, phase composition, and pore size. Such microspheres demonstrate photoprotective properties and can be used as cosmetic pigments.

## Figures and Tables

**Figure 1 materials-12-01472-f001:**
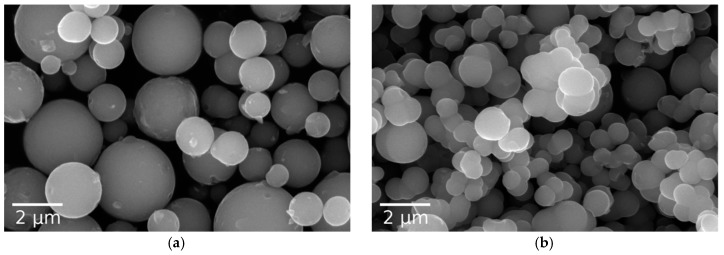
SEM images of the microspheres synthesized without the addition of NaOH, (**a**) *h* = 2 and (**b**) *h* = 6.

**Figure 2 materials-12-01472-f002:**
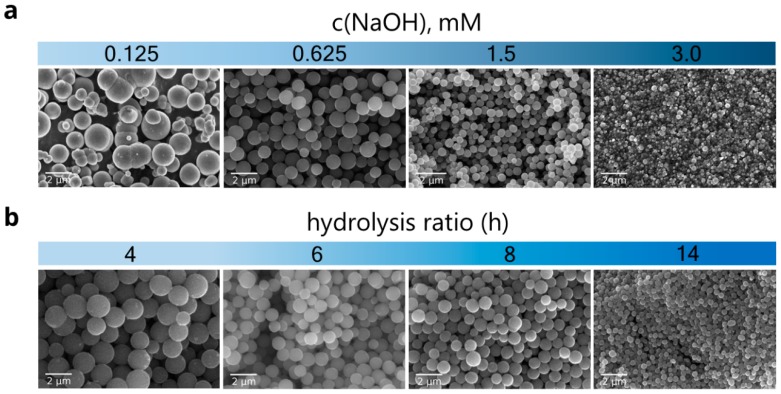
SEM images of the series of the samples that were prepared with varying NaOH concentration and a constant hydrolysis ratio of h = 6 (**a**); the samples prepared with varying hydrolysis ratios and a constant c(NaOH) = 0.625 mM (**b**).

**Figure 3 materials-12-01472-f003:**
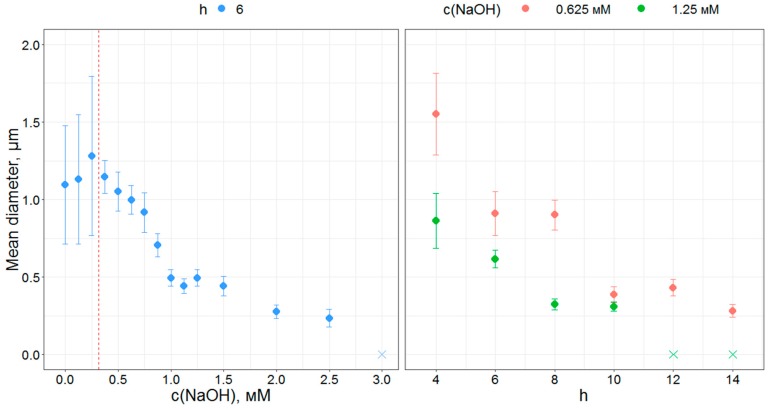
The changes of mean particle diameter and relative standard deviation according to SEM data with the change of NaOH concentration and hydrolysis ratio h. Crosses denote a lack of observed spherical particles.

**Figure 4 materials-12-01472-f004:**
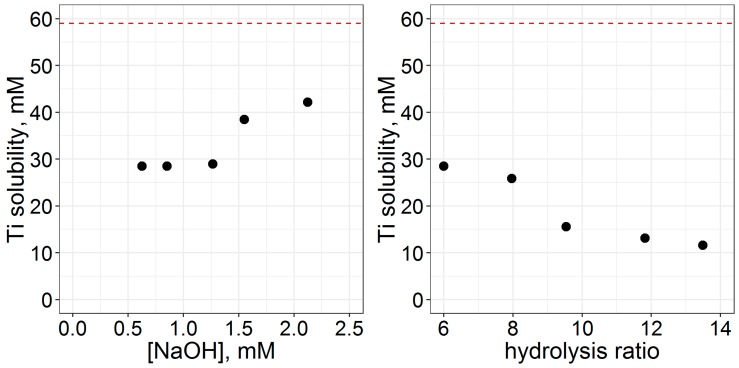
The concentration of Ti in mother liquors after two hours of the synthesis according to ICP-MS data; the dashed line denotes Ti(O^n^Bu)_4_ concentration in the beginning of the synthesis.

**Figure 5 materials-12-01472-f005:**
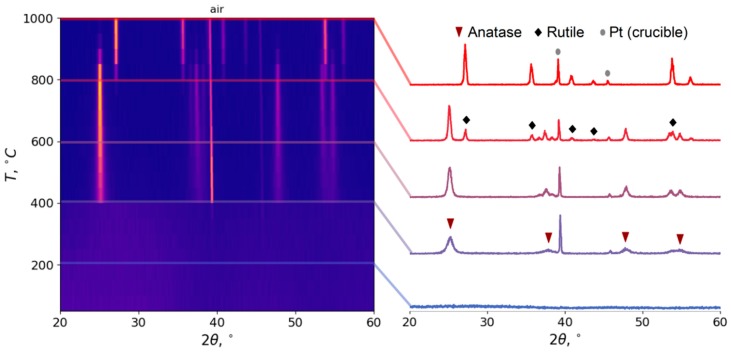
High-temperature X-ray diffraction (XRD) patterns of the spherical particles annealed in air; heating rate 5°/min.

**Figure 6 materials-12-01472-f006:**
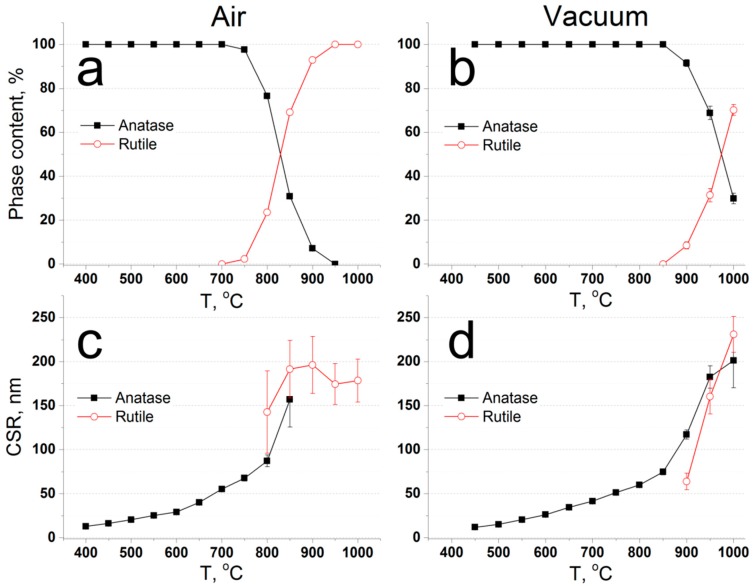
High-temperature XRD data: phase composition of the samples, annealed in air (**a**) and in vacuum (**b**), and the coherent scattering region sizes of the samples, annealed in air (**c**) and in vacuum (**d**).

**Figure 7 materials-12-01472-f007:**
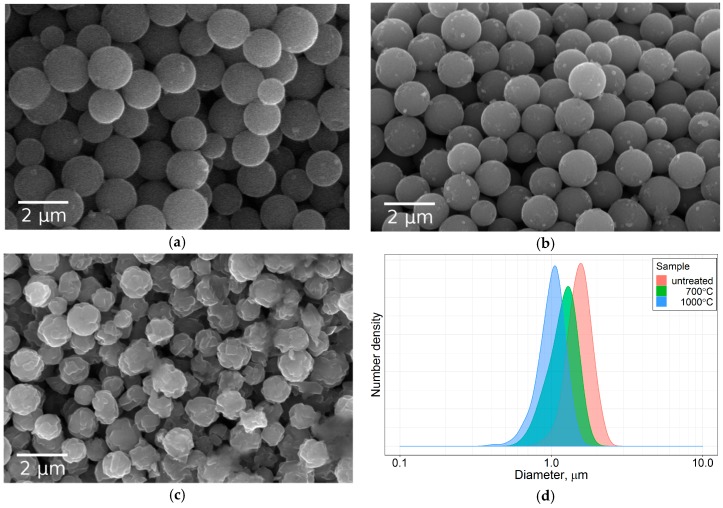
SEM shows that after annealing, the particles preserve their spherical morphology, but shrink in size: (**a**) untreated, (**b**) annealed at 700 °C, (**c**) 1000 °C, (**d**) respective particle size distributions according to analysis of SEM images.

**Figure 8 materials-12-01472-f008:**
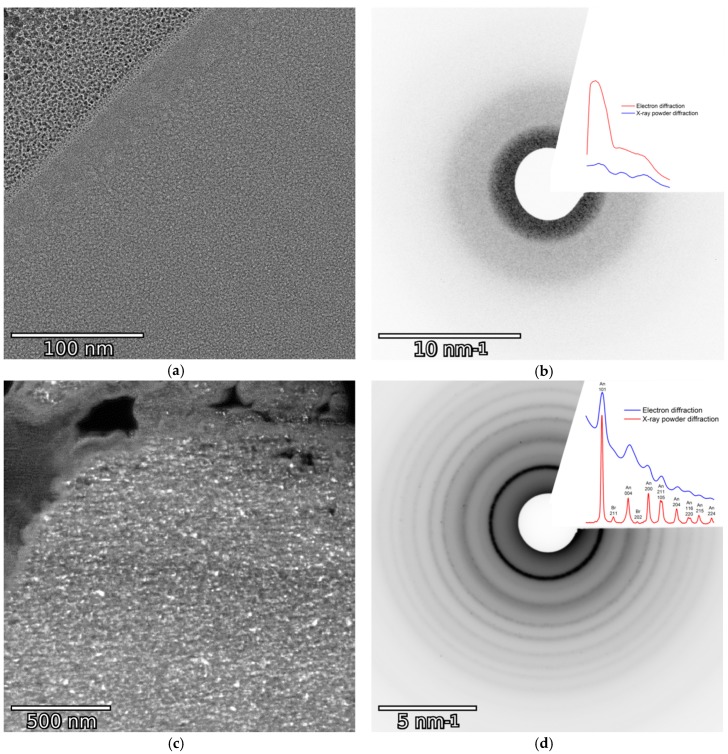
TEM image (**a**), ED and XRD (**b**) of the lamella cut out of the untreated spherical particle; dark-field TEM image (**c**), ED and XRD (**d**) of the lamella cut out of the particle hydrothermally treated at 200 °C for 24 h.

**Figure 9 materials-12-01472-f009:**
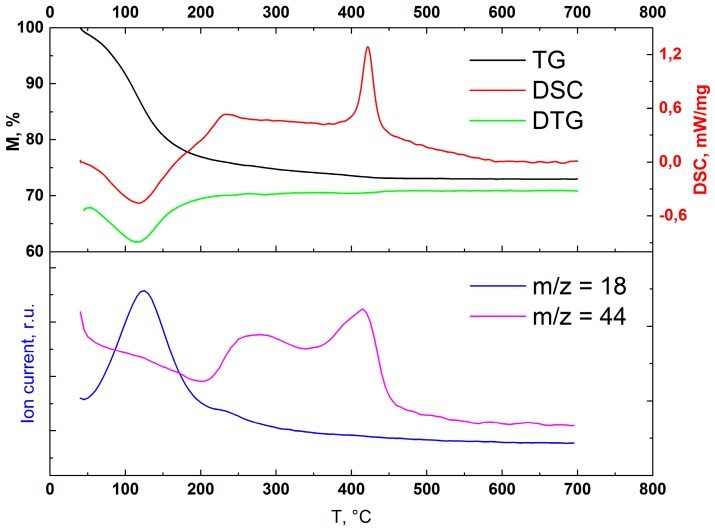
TG–DTG–DSC of untreated amorphous microspheres (upper picture) and MS-curves for evolved gases (lower picture). TG: thermogravimetry; DTG: derivative thermogravimetry; DSC: differential scanning calorimetry.

**Figure 10 materials-12-01472-f010:**
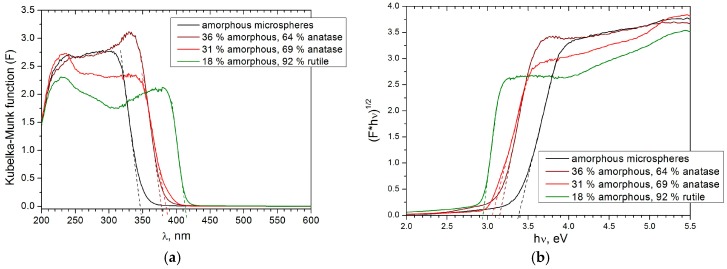
The optical absorption edges (**a**) and band-gap calculation (**b**) of amorphous, predominantly anatase, and predominantly rutile microspheres. The absorption edges of anatase-containing microspheres have similar positions, whereas for the rutile and amorphous phase, they differ significantly.

**Table 1 materials-12-01472-t001:** Phase composition, anatase coherent scattering region size (CSR), Brunauer–Emmett–Teller (BET) specific surface area, pore volume (Barrett–Joyner–Halenda, or BJH model), and average pore diameter of the obtained samples.

Sample Name	Phase Composition	CSR (anatase), nm	BET Surface Area, m^2^/g	V_pore_, cm^3^/g	D_pore_, nm
untreated	amorphous	-	220	0.10	2.6
an-400	anatase	24 ± 5	85	0.08	2.3
an-500	anatase	29 ± 4	65	0.13	4.9
an-625	anatase	45 ± 8	40	0.11	6.6
an-700	anatase	50 ± 11	20	0.08	7.9
ht-200-24	80 ± 2% anatase 20 ± 2% brookite	18 ± 1	115	0.34	9.8
ht-200-96	83 ± 2% anatase 17 ± 2% brookite	21 ± 1	80	0.33	9.8

**Table 2 materials-12-01472-t002:** Phase composition, optical band gap, and Urbach energy of amorphous, predominantly anatase, and predominantly rutile microspheres.

Phase Composition	Band Gap, eV	Urbach Energy, meV
amorphous	3.39	-
64% anatase, 36% amorphous	3.15	225
69% anatase, 31% amorphous	3.09	179
82% rutile, 18% amorphous	2.94	228

**Table 3 materials-12-01472-t003:** SPF of titanium oxide amorphous, predominantly anatase, and predominantly rutile microspheres and commercial pigment.

Phase Composition	Particle Mean Diameter, μm	SPF
amorphous	0.42	21
78% anatase, 22% amorphous	0.35	4
80% rutile, 20% amorphous	0.35	3
commercial cosmetic pigment (77% of rutile, 23% amorphous)	<0.2	8

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
