# Peer review of "Titanium Oxide Microspheres with Tunable Size and Phase Composition"

_materials, 2019, doi:10.3390/ma12091472_

Round 1
Reviewer 1 Report
MS No:
| materials-493933-peer-review-v1
|
Title:
| Titanium oxide microspheres with tunable size and phase composition
|
Authors:
| Anton S. Poluboyarinov, Vitaliy I. Chelpanov, Vasily A. Lebedev, Daniil A. Kozlov, Kristina A. Khazova, Dmitry S. Volkov, Irina V. Kolesnik, Alexey A. Garshev |
This paper deals with the synthesis and characterization of titanium oxide microspheres that can be used in dye-sensitized solar cells and as sorbent materials for high performance liquid chromatography. Both preparation method and physicochemical characterization are presented in detail. A wide variety of characterization methods are used in order to correlate preparation conditions with particle size, phase composition and porosity of the final products.
Overall this paper is well-written and organized providing useful information. However, a major point should be addressed before publication on Materials. In specific, I would advise the authors to provide a simple application showing the activity of these materials as absorbents; this is important in order to provide a correlation between TiO2 microspheres activity and their physicochemical properties, pointing out the significance of the present work.
Author Response
Point 1:
a major point should be addressed before publication on Materials. In specific, I would advise the authors to provide a simple application showing the activity of these materials as absorbents; this is important in order to provide a correlation between TiO2 microspheres activity and their physicochemical properties, pointing out the significance of the present work.
Response 1:
Dear Sir or Madam,
thank you for your comments and recommendations. We have updated our paper and provided an application showing properties of the microspheres as materials. Unfortunately we cannot provide the properties of the microsperes as sorbents because these data have been already published (Refs. 15 and 20 in the manuscript). We added to the manuscript a demonstration of optical and photoprotective properties of the microspheres. SPF factors of TiO2 microspheres have not been measured before and we believe that this is an interesting demonstration of their functional properties.
Reviewer 2 Report
The authors describe the synthesis of spherical titania particles with controllable size, crystallinity and pore size that can be used in dye-sensitized solar cells and as sorbent materials for high performance liquid chromatography.
The study is interesting and the results appreciable.
However some poins should be added.
In the introduction I suggest the updating of references and also the description of the use of TiO2 in the photocatalytic applications as in the paper Scientific Reports (2015) 5:17801.
To provide useful details to comunity working in the field of DSSCs and photocathic materials it is important to have an estimation of the band gaps in the different nanoparticles.
To this purpose, the band gap calculation is possible by Tauc plot or Kubleka-Munk method, in the first case by using absorbance and in the second case by using Reflectance, in both case the calculation of band gap correspond to the tangent intercept of the plot on the x axis(eV).
For the band gap calculation the autors should refer to these papers: Applied Surface Science 441 (2018) 575–587; Superlattices and Microstructures 89 (2016) 153 169.
Author Response
Point 1:
In the introduction I suggest the updating of references and also the description of the use of TiO2 in the photocatalytic applications as in the paper Scientific Reports (2015) 5:17801.
Response 1:
Dear Sir or Madam, thank you for your comments and recommendations. We updated references and added recent works including the use of TiO2 microspheres for actinide photocatalytic reduction (Ref. 20). However, we avoided detailed discussion of photocatalysis on TiO2 on the Introduction because we did not discuss photocatalytic activity of the microspheres in this paper.
Point 2:
To provide useful details to comunity working in the field of DSSCs and photocathic materials it is important to have an estimation of the band gaps in the different nanoparticles.
To this purpose, the band gap calculation is possible by Tauc plot or Kubleka-Munk method, in the first case by using absorbance and in the second case by using Reflectance, in both case the calculation of band gap correspond to the tangent intercept of the plot on the x axis(eV).
For the band gap calculation the autors should refer to these papers: Applied Surface Science 441 (2018) 575–587; Superlattices and Microstructures 89 (2016) 153 169.
Response 2:
We recordered UV-Vis specrta in diffuse reflectance mode because our samples are in powder form. Optical band gap and Urbach energy for amorphous, anatase and rutile microspheres was calculated from the spectra. Band gap values of anatase and rutile microspheres are close to literature data. The results have been added to the manuscript.
Round 2
Reviewer 1 Report
Accept
Author Response
Point 1
English language and style are fine/minor spell check required
Response:
MDPI editing service will be used to improve English.
Reviewer 2 Report
About the response 2:
The band gap calculation is possible by Tauc plot or Kubleka-Munk methods also if the material is in the power form as reported in the references previously suggested.
Therefore I confirm the request about this calculation in order also to compare the results. Are the results different? Why? A discussion about the obtained results results can be apported.
Author Response
Point 1
The band gap calculation is possible by Tauc plot or Kubleka-Munk methods also if the material is in the power form as reported in the references previously suggested.
Therefore I confirm the request about this calculation in order also to compare the results. Are the results different? Why? A discussion about the obtained results results can be apported.
Response 1
The calculation of band gap using Kubelka-Munk function and discussion were added to the manuscript. The band gap of rutile is close to literature data. Anatase band gap in literature varies from 3.05 to 3.2 eV, probably because of the defect states and/or presence of the amorphous phase. Band gap of the microspheres prepared in frames of this work is 3.09 and 3.15 eV. The band gap for amorphous titanium oxide was also determined and equals 3.39 eV. This value correlates with the band gap of very small (1 nm) anatase particles.